# Identification and Characterization of a New Regulator, TagR, for Environmental Stress Resistance Based on the DNA Methylome of *Streptomyces roseosporus*

Wen-Li Gao,[a,b] Jiao-Le Fang,[a,b] Chen-Yang Zhu,[a,b] Wei-Feng Xu,[a,b] Zhong-Yuan Lyu,[a,b] Xin-Ai Chan,[a,b] Qing-Wei Zhao,[a]
Yong-Quan Li[a,b]

aFirst Affiliated Hospital and Institute of Pharmaceutical Biotechnology, Zhejiang University School of Medicine, Hangzhou, China
bZhejiang Provincial Key Laboratory for Microbial Biochemistry and Metabolic Engineering, Institute of Pharmaceutical Biotechnology, Hangzhou, China

**ABSTRACT** DNA methylation is a defense that microorganisms use against extreme environmental stress, and improving resistance against environmental stress is essential for industrial actinomycetes. However, research on strain optimization utilizing DNA methylation for breakthroughs is rare. Based on DNA methylome analysis and KEGG pathway assignment in *Streptomyces roseosporus*, we discovered an environmental stress resistance regulator, TagR. A series of *in vivo* and *in vitro* experiments identified TagR as a negative regulator, and it is the first reported regulator of the wall teichoic acid (WTA) ABC transport system. Further study showed that TagR had a positive self-regulatory loop and m4C methylation in the promoter improved its expression. The Δ*tagR* mutant exhibited better hyperosmotic resistance and higher decanoic acid tolerance than the wild type, which led to a 100% increase in the yield of daptomycin. Moreover, enhancing the expression of the WTA transporter resulted in better osmotic stress resistance in *Streptomyces lividans* TK24, indicating the potential for wide application of the TagR-WTA transporter regulatory pathway. This research confirmed the feasibility and effectiveness of mining regulators of environmental stress resistance based on the DNA methylome, characterized the mechanism of TagR, and improved the resistance and daptomycin yield of strains. Furthermore, this research provides a new perspective on the optimization of industrial actinomycetes.

**IMPORTANCE** This study established a novel strategy for screening regulators of environmental stress resistance based on the DNA methylome and discovered a new regulator, TagR. The TagR-WTA transporter regulatory pathway improved the resistance and antibiotic yield of strains and has the potential for wide application. Our research provides a new perspective on the optimization and reconstruction of industrial actinomycetes.

**KEYWORDS** DNA methylome, environmental stress resistance regulator, TagR, WTA ABC transport system, daptomycin, *Streptomyces rosesosporus*

Address correspondence to Yong-Quan Li, lyq@zju.edu.cn, or Qing-Wei Zhao, qwzhao@zju.edu.cn.
The authors declare no conflict of interest.
*[This article was published on 8 May 2023 with an incorrect grant in Acknowledgments. The Acknowledgments were corrected in the current version, posted on 15 June 2023.]*

Actinomycetes are the main industrial producer of microbial drugs (more than 60%). Many microorganisms, including actinomycetes, generally face environmental stresses during fermentation, which considerably affects microbial physiology (1). Environmental stresses such as free fatty acids added as precursors and hyperosmotic stress caused by metabolites are sometimes unavoidable (1–3). Hyperosmotic stress can concentrate biomolecules, hinder biofilm formation, and inhibit nutrient transport (4–6). Extracellular free fatty acids can disrupt the electron transport chain, interfere with oxidative phosphorylation, and destroy the phospholipid bilayer (7, 8). Therefore, increasing the environmental adaptability of strains and particularly their resistance to hyperosmotic stress and free fatty acids can effectively increase the productivity of strains.

Microorganisms have developed numerous mechanisms to resist environmental stress. Among them, epigenetic modification is a low-cost and low-risk strategy that avoids the

accumulation of harmful mutations and simultaneously allows the adaptation to environmental challenges (9–12). As the most common epigenetic modification, DNA methylation includes 3 main modifications: m6A, m5C, and m4C (13, 14). DNA methylation not only helps proteins recognize alien DNA fragments, such as Sco5333 from *Streptomyces coelicolor* and Tbis1 from *Thermobispora bispora* recognize m5C (15), but also regulates gene expression. The phase-variable modification of regulons achieved by m6A DNA methyltransferase has broad functions in changing microbial phenotypes to enable adaptation to the environment (16–18). For example, DNA adenine methylase (Dam) can coordinate with the regulator HdfR to regulate the expression of the chaperone-usher fimbriae *std* operon in *Salmonella enterica* and with the regulator Fur to regulate the *sci*1 T6SS cluster in *Escherichia coli* (19, 20). The m5C and m4C modifications can also directly or indirectly regulate the interaction between bacteria and the environment. Cytosine methylation controls morphophysiological differentiation and actinorhodin production in *S. coelicolor* (21). In *Leptospira interrogans*, m4C methyltransferase is critical for virulence and can regulate the expression of the extracytoplasmic function sigma factor (22–25). The interaction between DNA methylation and transcriptional regulators helps strains resist environmental stress (26, 27). The regulation of methylation provides a valuable resource for mining regulators of environmental stress resistance. However, few studies have considered this resource or optimized industrial actinomycetes using methylation to enable breakthroughs in this area.

*Streptomyces roseosporus* is the industrial producer of daptomycin, which is an important antibiotic used against infections caused by drug-resistant pathogens (28, 29). The addition of decanoic acid (DA) to the culture broth was shown to be essential for increasing daptomycin yield and productivity(30). However, as a long-chain free fatty acid, DA is toxic, and large amounts of DA can cause cell death (8, 31). In addition, metabolite accumulation during the fermentation process results in a hypertonic environment. To address these disadvantages during fermentation, research on environmental stress resistance should investigate a new perspective. Our previous study identified a new DNA methyltransferase, SroLm3, which plays a global regulatory role in *S. roseosporus* (23). Therefore, *S. roseosporus* was chosen as the research object to screen for regulatory factors and optimize environmental stress resistance.

This study established a strategy for mining environmental stress resistance regulators based on the DNA methylome in *S. roseosporus* and discovered a new regulator of the wall teichoic acid (WTA) ABC transport system, TagR. Further analysis revealed the mechanisms of TagR involvement, leading to its application to improve strain resistance and the yield of daptomycin. Our research provides a new perspective on the reconstruction of industrial strains.

## RESULTS

**Mining the regulator gene of environmental stress resistance *tagR*.** SroLm3, an m4C DNA methyltransferase in *S. roseosporus*, is responsible for the global regulation of secondary metabolism. Whole-genome sequencing and single molecule real-time (SMRT) sequencing of *S. roseosporus* L30 (wild type [WT]) and *S. roseosporus* L33 (Δ*sroLm3*) were conducted in a previous study (23). Herein, the specific strategy applied was to screen for regulatory genes based on the difference in methylation sites between WT and Δ*sroLm3*; furthermore, the abundance of genes assigned to the KEGG environmental information processing pathway was comprehensively considered. The chromosomal distribution of modification sites is shown in Fig. 1A. There were 23,847 m4C modification sites in WT and 15,646 in Δ*sroLm3*, and there were 552 presumed regulatory genes in the genome. Most of the modifications were in the coding DNA sequence (CDS) region, and therefore, regulatory genes were screened according to the modification differences in this region. We first identified regulatory genes with more than 80% loss of m4C methylation sites and more than 1% loss of m4C methylation abundance. Seven genes were identified and are listed in Table 1.

Then, regions with a high density of genes in environmental information processing

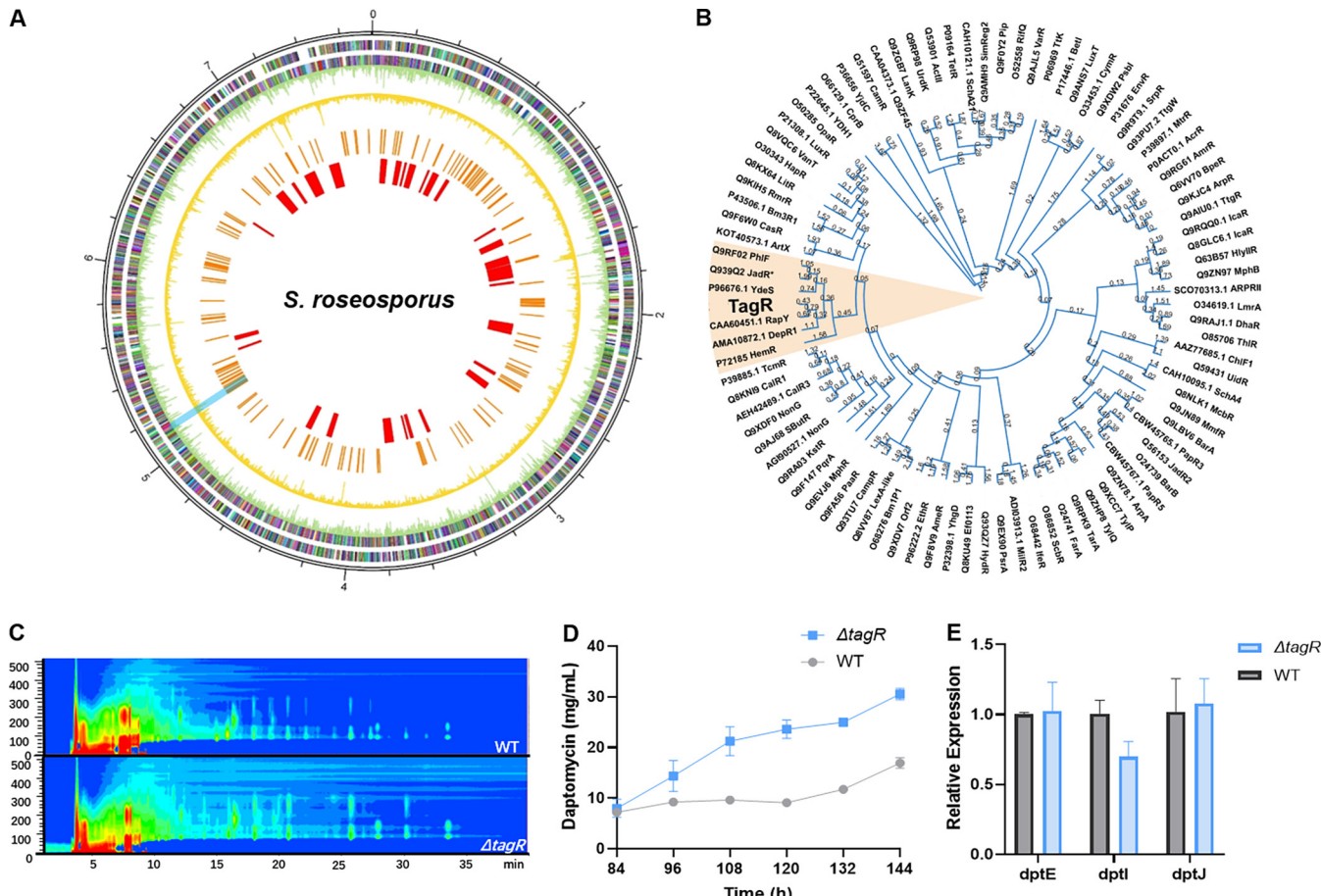

**FIG 1** Schematic representation of the genome, phylogenetic tree, and fermentation yield of Δ*tagR*. (A) Schematic representation of the distribution of methylation sites and environmental information processing pathway genes in the genome. From the inner circle to the outer circle, biosynthesis gene clusters (BGC), KEGG environmental information processing pathway genes, WT m4C methylome, Δ*sroLm3* m4C methylome, forward strand CDS, and reverse strand CDS are listed in sequence. The cyan line shows the position of *tagR*. (B) Phylogenetic tree of partial TetR family transcriptional regulators. (C) UV absorption spectra (215 nm) of the fermentation broths of WT and Δ*tagR*. (D) Yields of daptomycin from the WT and Δ*tagR* fermentation experiments (*n* = 3, mean with SD). (E) Relative expression of the daptomycin gene cluster in WT and Δ*tagR* (*n* = 3, mean with SD).

pathways were identified by KEGG analysis. Regulatory genes in these regions are more likely to be related to the regulation of environmental stress resistance. A total of 7,144 open reading frames were predicted on the chromosome, among which 269 genes were assigned to the environmental information processing pathway. The regions with a high density of these genes (more than 10%) were *orf700-orf900*, *orf1600-orf1700*, *orf2200-orf2300*, and *orf4700-orf4900*, which are shown in Fig. S1A and marked in blue. Of the seven genes in Table 1, only *tagR* (*orf4759*, named according to the functional annotation and the results of subsequent studies) and *orf4820* were in these regions. We noted that *tagR* (indicated by a cyan rectangle in Fig. 1A) had a 100% loss of m4C methylation in both the CDS

**TABLE 1** Regulatory genes identified by methylome analysis

| No. | Gene | Loss of CDS m4C methylation sites | Loss of m4C methylation abundance[a] | Loss of promoter m4C methylation | Function |
|---|---|---|---|---|---|
| 1 | *orf199* | 86% (6/7) | 1% | 0 | Transcriptional regulator, AcrR family |
| 2 | *orf441* | 88% (7/8) | 1.3% | 0 | Regulatory protein |
| 3 | *orf1070* | 83% (20/24) | 1.1% | 0 | Regulator of polyketide synthase expression |
| 4 | *orf1818* | 92% (12/13) | 1% | 0 | Transcriptional regulator, CdaR-family |
| 5 | *orf2391* | 83% (5/6) | 1.3% | 0 | Transcriptional regulator, WhiB family |
| 6 | *tagR* (*orf4759*) | 100% (9/9) | 1.3% | 100% (1/1) | Transcriptional regulator, TetR family |
| 7 | *orf4820* | 92% (11/12) | 1.7% | 0 | Transcriptional regulator, IclR family |

[a]Methylation abundance is the number of CDS m4C-methylated sites/CDS base number.

and promoter region, which was higher than that in *orf4820*. The length of *tagR* was found to be 621 bp, and it encoded 206 amino acids. In WT, cytosine was modified at 9 bp upstream of *tagR*, and there were 9 cytosine bases modified in the CDS region, which were at positions 213, 237, 309, 314, 423, 462, 476, 477, and 554. All these modifications disappeared in the Δ*sroLm3* mutant. Therefore, *tagR* was chosen as the typical example of our mining strategy for further research.

TagR was annotated as a TetR family protein. The upstream gene of *tagR* was annotated as an ABC transporter, and its downstream gene was a hypothetical protein, both of which had different transcriptional directions from *tagR*. TagR has typical TetR structures, including an N-terminal HTH DNA-binding motif and a C-terminal ligand recognition domain. Studies involving multiple TetR family regulators indicated that these proteins might act as environmental information sensors that regulate gene expression in response to various stimuli (32–34); this conclusion is consistent with the characteristics of our target regulator. A total of 90 TetR family proteins with functions that are listed in the NCBI protein database were chosen to generate a phylogenetic tree (Fig. 1B). According to the phylogenetic tree, TagR had the highest homology with *Streptomyces rapamycinicus* RapY, a pathway-specific regulator of rapamycin biosynthesis (35), with more than 40% protein identity. Another protein on the same branch, DepR1, which is a regulator in *S. roseosporus*, is a direct positive regulator of daptomycin biosynthesis (36). Therefore, a series of experiments were performed to explore the regulatory role of TagR in daptomycin production.

The plasmid pKC1139-Δ*tagR* was used as the knockout vector, and *tagR* was deleted in-frame in the WT. Genotype verification of *S. roseosporus* Δ*tagR* is provided in the supplemental material (Fig. S2A and B). WT and Δ*tagR* were cultured in yeast extract-malt extract (YEME) medium for 144 h. There was no significant difference in the biomass between the two strains during fermentation (Fig. S1C). Moreover, the fermentation broth was analyzed by high-performance liquid chromatography (HPLC). Their UV absorption spectrum (215 nm) showed an overall enhancement in the metabolite levels (Fig. 1C). HPLC analysis also showed a difference in the daptomycin yield between the Δ*tagR* mutant and WT at 144 h (Fig. S1B). The yield curve showed that the accumulation of daptomycin (slope) in the Δ*tagR* was faster than that in the WT during the early stage (Fig. 1D). At the end of fermentation, the daptomycin yield of the Δ*tagR* was 70% higher than that of the WT. The *tagR* complementation experiment confirmed that the yield change in the Δ*tagR* mutant was caused by *tagR* deletion (Fig. S3A and C). To investigate the mechanism underlying the increased yield of daptomycin, the expression of the major biosynthetic genes of daptomycin in the Δ*tagR* mutant and WT was analyzed (Fig. S1D, key promoters are indicated with black arrows). Surprisingly, no significant increase was found in the expression levels of the major biosynthetic genes of daptomycin (Fig. 1E), indicating that the increased yield of daptomycin observed in the Δ*tagR* mutant was not induced by higher expression levels of biosynthetic genes. The mechanism underlying the increase in the yield of daptomycin and the relationship between the regulator TagR and environmental stress resistance were the next topics that we addressed in our study.

**Analysis of *tagR* function by the Δ*tagR* transcriptome sequencing.** To investigate the mechanism underlying the increased daptomycin production and to identify the function of TagR, transcriptome sequencing was performed in the Δ*tagR* mutant and WT. Based on the growth curves, transcriptome sequencing was performed at 48 h (log phase), 72 h (late log phase), and 120 h (stationary growth phase). The chromosomal distribution of differentially expressed genes between Δ*tagR* and WT is shown in Fig. 2A. There was no significant difference in the expression of the daptomycin synthetic gene cluster, which was consistent with previous results (Fig. S4A). More than 6,000 genes were detected at each time point, but no more than 10% of the genes had different expression levels, and in some samples, there were as few as 2% (Fig. S4B). These results suggest that TagR is more likely to be a pathway-specific regulator than a global regulator. KEGG pathway enrichment analysis showed that the differentially expressed genes at each time point, especially at 48 h and 120 h, were enriched

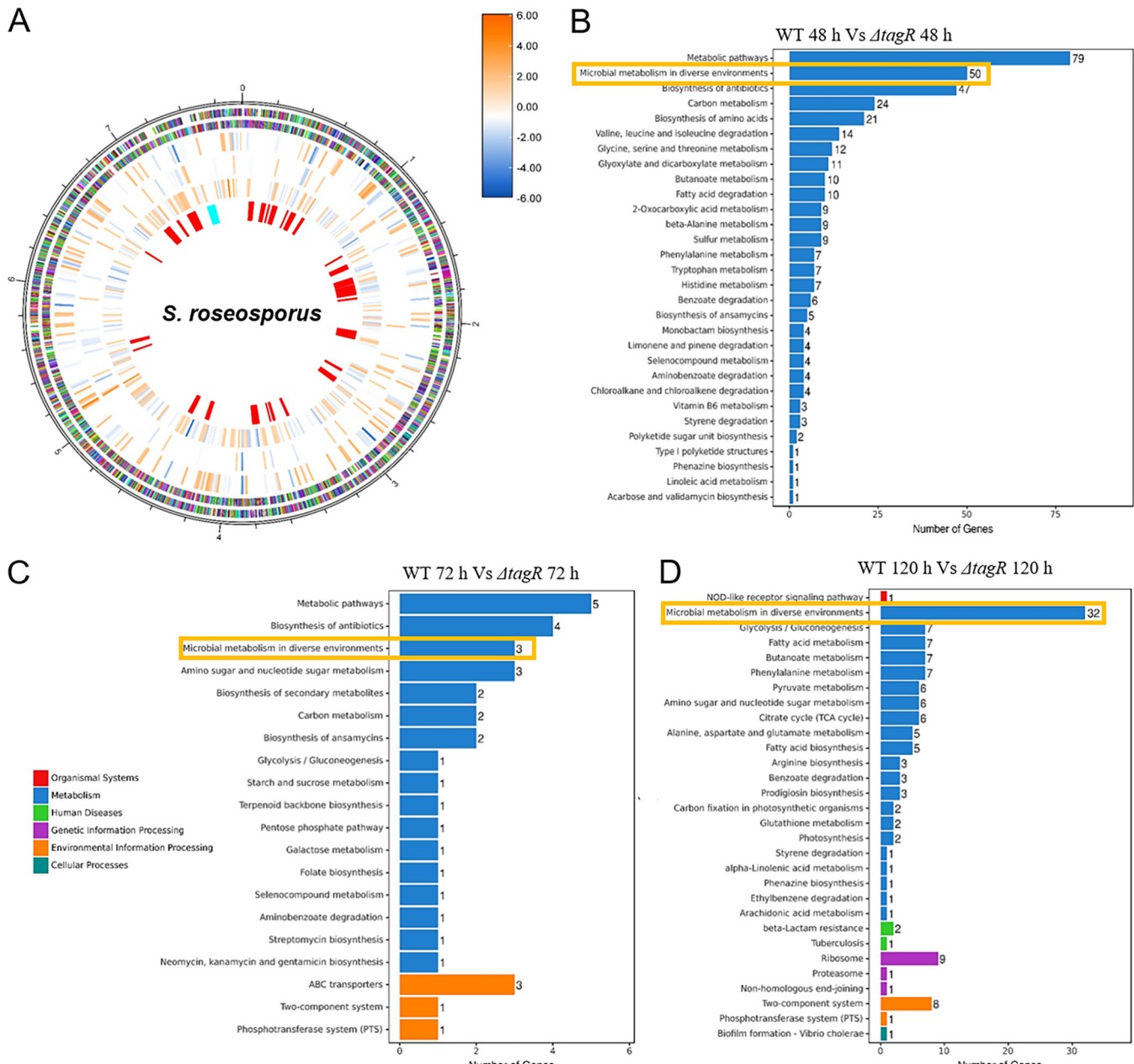

**FIG 2** WT and Δ*tagR* transcriptomes (48, 72, and 120 h). (A) Map of the distribution of differentially expressed genes in the genome. From the inner circle to the outer circle, BGC, differentially expressed genes at 48 h, differentially expressed genes at 72 h, differentially expressed genes at 120 h, the plus chain CDS, and anti-chain CDS are listed in sequence. (B to D) Distribution statistics of differential gene enrichment in KEGG pathways in WT and Δ*tagR* transcriptomes (48, 72, and 120 h).

in the microbial metabolism in diverse environments pathway. Accordingly, TagR might regulate strain adaptation to environmental stress (Fig. 2B to D).

By analyzing the transcriptomes of Δ*tagR* and WT, we identified 12 crossover genes among the differentially expressed genes at the 3 time points (Fig. S4B); the annotation information is shown in Table 2. The expression levels of the genes in lines 1 and 2, named *tagH* and *tagG* according to the annotation and the results of subsequent studies, were upregulated at all 3 time points; moreover, the expression of the genes in lines 3 to 9 was downregulated. However, the expression levels of the remaining 3 genes did not change in the same direction at the 3 time points, which implied that these 3 genes in lines 10, 11, and 12 might not be directly regulated by TagR. Therefore, the former 9 genes were selected for further study to investigate the regulatory mechanisms of TagR.

**TABLE 2** Differentially expressed genes in WT and ΔtagR strain transcriptomes (intersection of all time points)

| No. | Gene | Product | Log$_2$-fold change at 48 h | Log$_2$-fold change at 72 h | Log$_2$-fold change at 120 h |
|---|---|---|---|---|---|
| 1 | tagH(orf4757) | ABC transporter, ATP-binding protein | 6.06 | 6.19 | 6.24 |
| 2 | tagG(orf4758) | O-antigen export system, permease protein | 4.72 | 5.01 | 4.48 |
| 3 | orf488 | Arsenical resistance operon repressor | −1.26 | −2.30 | −4.24 |
| 4 | orf492 | Flavin-dependent monooxygenase ArsO associated with arsenic resistance | −1.33 | −1.92 | −2.17 |
| 5 | orf494 | Arsenical resistance protein ACR3 | −2.07 | −2.61 | −2.43 |
| 6 | orf495 | Arsenical resistance operon repressor | −2.41 | −2.86 | −1.76 |
| 7 | orf496 | Arsenate reductase (EC 1.20.4.4) thioredoxin-coupled, LMWP family | −2.15 | −2.27 | −1.76 |
| 8 | orf497 | Thioredoxin reductase (EC 1.8.1.9) | −1.80 | −1.85 | −1.94 |
| 9 | orf3562 | Arsenical resistance protein ACR3 | −2.43 | −3.83 | −4.30 |
| 10 | orf1359 | NADH:flavin oxidoreductase/NADH oxidase | −2.08 | −1.18 | 1.93 |
| 11 | orf2787 | Peptidase S1 and S6, chymotrypsin/Hap | −2.85 | 1.21 | 3.19 |
| 12 | orf961 | Hypothetical protein | −1.48 | 1.17 | −3.46 |

**Regulatory mechanism of *tagR*.** To elucidate the regulatory mechanism of TagR, we investigated its binding sites, as well as its possible self-regulation and DNA methylation-based regulatory processes.

TetR family transcription factors can regulate transcription by binding to the promoter regions of their target genes (37). The binding of TagR to the promoter region of the genes listed in Table 2 (lines 1 to 9) was examined using the electrophoretic mobility shift assay (EMSA). The results showed that TagR was specifically bound to $P_{tagR(tagG)}$ (Fig. 3A), indicating that TagR can directly regulate *tagR* and upstream cotranscription genes *tagG* and *tagH*. It is worth noting that there were multiple TagR-$P_{tagR(tagG)}$ bands, which indicated that TagR might have multiple binding sites on this probe. No binding was detected in the other promoter fragments (Fig. S5B and C), suggesting indirect regulation. Transcriptome analysis showed that the expression of *tagG* and *tagH* was significantly upregulated after *tagR* knockout, indicating that TagR negatively regulates both genes.

The DNA footprinting assay results showed that TagR had at least two binding sites on $P_{tagR(tagG)}$ (Fig. 3B and C); the detected binding sites are indicated in bold and underlined fonts. Previous studies of members of the TetR family have revealed that they usually function as dimers (38, 39) and have palindromic motifs at their binding sites (32, 37). A palindromic motif was found in the sequence of binding site 1 (highlighted in yellow in Fig. 3C), which was consistent with the general characteristics of this family. To further determine the protein binding site, we performed mutation experiments (10 ± 1 bp per mutation) based on the results of DNA footprinting. Mutation experiments confirmed that the protein did not bind DNA only when all the bases indicated in red in Fig. 3C were mutated (Fig. S5D). Thus, we identified the binding sequences of TagR on $P_{tagR(tagG)}$.

Based on the methylome data, there was an m4C methylation modification on $P_{tagR}$ in the WT (Fig. 3C, highlighted in purple), which was absent in Δ*sroLm3*. To evaluate the possible self-regulation and DNA methylation-based regulation of TagR, the *neo* (kanamycin resistance gene) and *gusA* reporter genes were expressed in tandem, and their shared promoters were replaced with $P_{tagR}$ to generate the GusA reporter system (Fig. S6). The plasmid pRM02-*neo*-*gusA*-$P_{tagR}$ was conjugated into WT, Δ*tagR*, and Δ*sroLm3*Δ*tagR* strains to generate WT/*gusA*, Δ*tagR*/*gusA*, and Δ*sroLm3*Δ*tagR*/*gusA* strains. By comparing the relative enzymatic activities of GusA in WT/*gusA*, Δ*tagR*/*gusA*, and Δ*sroLm3*Δ*tagR*/*gusA*, we determined how TagR and methylation affected the strength of $P_{tagR}$. The strains were cultured in tryptic soy broth (TSB) and YEME for 36 h, and mycelia were collected for spectrophotometric measurement of glucuronidase activity. The relative enzyme activity of GusA was lower in the Δ*tagR*/*gusA* strain than in the WT/*gusA* strain (Fig. 3D), indicating that TagR can positively regulate its own promoter $P_{tagR}$, forming a positive self-regulatory loop. The relative enzyme activity of the Δ*sroLm3*Δ*tagR*/*gusA* strain was lower than that of the Δ*tagR*/*gusA* strain (Fig. 3D), indicating that SroLm3 had a positive regulatory effect on $P_{tagR}$ (see Fig. 5A). However, it is not clear whether this regulation is direct or indirect.

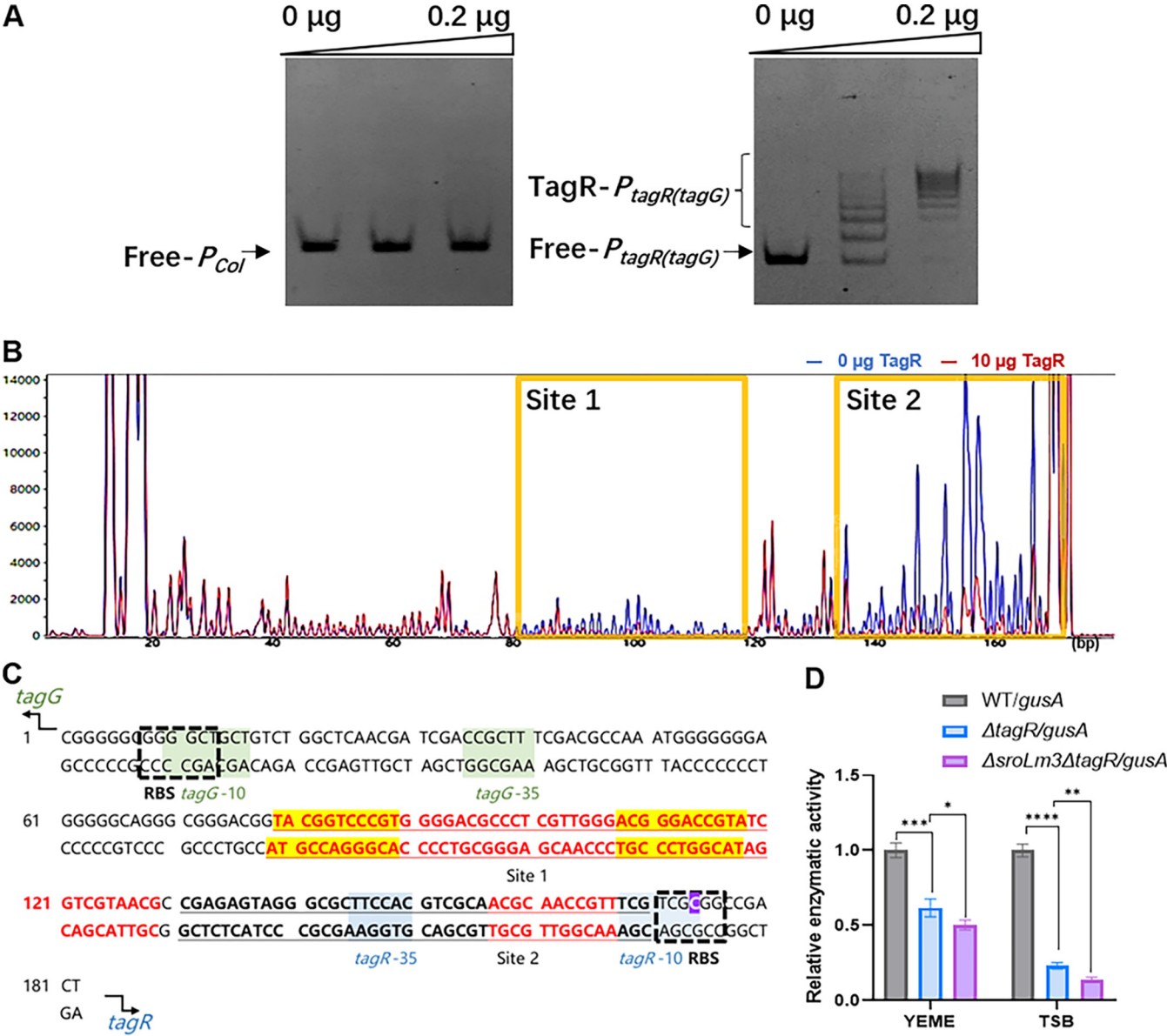

**FIG 3** Binding site assays of TagR. (A) EMSA. (B) DNA footprinting assay. The blue chromatogram line is the group without TagR, and the red chromatogram line is the group with the addition of 10 $\mu$g of TagR. The binding sites are marked in gold. (C) Binding sites verified by mutation (marked in scarlet). The footprinting assay result is indicated in bold and underlined font. The palindrome motif is highlighted in yellow. The m4C methylation site is highlighted in purple. The putative −10 and −35 sites of *tagR* are marked in blue, and the putative −10 and −35 sites of *tagG* are marked in green. The putative ribosome-binding sites (RBSs) are circled with a black dotted line. (D) GusA reporter gene assay to compare P$_{tagR(tagG)}$ strength in WT, Δ*tagR*, and Δ*sroLm3*Δ*tagR* (n = 3, mean with SD). $P < 0.05$ was summarized with 1 asterisk [*], $P < 0.01$ was summarized with 2 asterisks [**]; $P < 0.001$ was summarized with 3 asterisks [***]; $P < 0.0001$ was summarized with 4 asterisks [****].

**Functions of the TagR target genes *tagG* and *tagH*.** To determine the specific pathways regulated by TagR, the functions of its target genes, *S. roseosporus tagG* (*SrtagG*) and *tagH* (*SrtagH*), were studied. As the same functional genes were described in different species, we added the species abbreviation before the protein or gene name when necessary for emphasis. For example, *S. roseosporus tagG* is referred to as *SrtagG*, and the protein is referred to as SrTagG, while *Bacillus subtilis tagG* is referred to as *BstagG*. According to NCBI BLAST, the genes *SrtagG* and *SrtagH* were predicted to encode an ABC transport system TagG superfamily permease and TagH superfamily ATP-binding protein, respectively. Based on the NCBI Protein Data Bank BLAST, there were two proteins with more than 40% identity with SrTagH. SrTagH was homologous to the N-terminal domain of *B. subtilis* TagH (BsTagH-N) (PDB 7DD0_A), with 46.57%

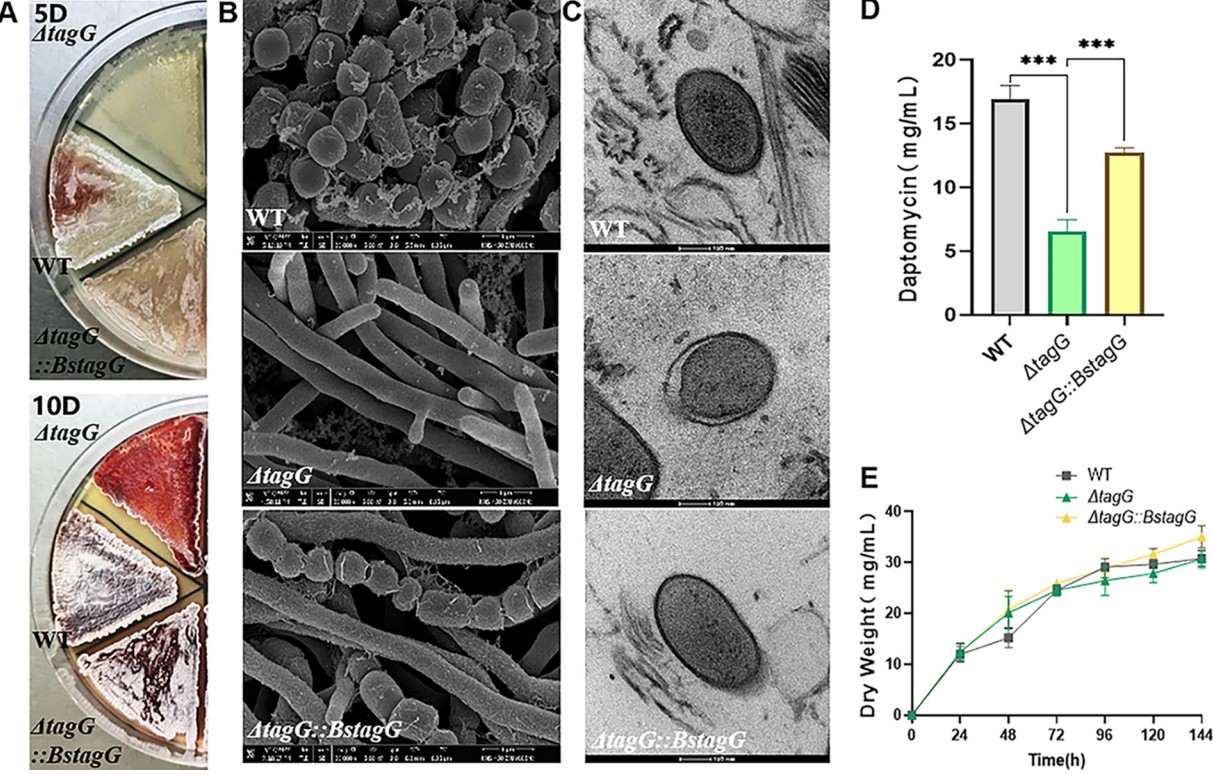

**FIG 4** Morphological diversities and fermentation yields of WT, ΔtagG, and ΔtagG::BstagG. (A) Growth status of the WT, ΔtagG, and ΔtagG::BstagG strains. (B) SEM images of WT, ΔtagG, and ΔtagG::BstagG (bar= 1 μm). (C) TEM images of the WT, ΔtagG, and ΔtagG::BstagG (bar= 100 nm). (D) Yields of daptomycin from WT, ΔtagG, and ΔtagG::BstagG fermentation experiments ($n$ = 3, mean with SD). (E) Growth curves of WT, ΔtagG, and ΔtagG::BstagG in fermentation experiments ($n$ = 3, mean with SD). $P < 0.001$ was summarized with 3 asterisks [***].

identity, and to chain A of *Alicyclobacillus herbarius* TarH (AhTarH-A) (PDB 6JBH_A), with 42.86% identity (Fig. S7A). BsTagH-N and AhTarH have been identified as ATP-binding proteins of the WTA ABC transport system (40, 41). Based on the annotation and homology alignment, the function of SrTagH is probably related to the transport of WTA, and we speculate that its cotranscriptional gene-encoded protein, SrtagG, may also involve in WTA transport. The only homologous protein of SrTagG in the Protein Data Bank was chain C of *A. herbarius* TarG (AhTarG-C) (PDB 6JBH_C), with 23.11% homology (Fig. S7B). The homology between SrTagG and *B. subtilis* TagG (BsTagG) (NP_391452.1) was 19.70%. AhTarG-C and BsTagG are adjacent to AhTarG-A and BsTagH in the genomes, respectively, and they both function as permeases of the WTA ABC transport system (40, 42).

Heterologous functional complementation was used to further study the function of *SrtagG*. *SrtagG* was knocked out, and the genotype of the *S. roseosporus* ΔtagG was verified by PCR (Fig. S2A and C). According to previous studies, the WTA transporter permease and ATP-binding protein cooperate to flip WTA from the cytoplasm to the extracellular space (39). Thus, the knockout of the predicted permease *SrtagG* may result in a range of WTA-deficient phenotypes in *S. roseosporus*. It has been reported that WTA deficiency delays growth and affects differentiation in *S. coelicolor* (43). In addition, WTA plays an important role in the adaptation to environmental ion concentrations, and its deficiency leads to decreased adaptability (44, 45). The results of culture experiments showed that the growth of ΔtagG was slowed, and spore differentiation was delayed and significantly reduced (Fig. 4A). No spores were observed by scanning electron microscopy (SEM) imaging of the ΔtagG strain, and its mycelial surface was smooth, while that of the WT was rough (Fig. 4B). Transmission electron microscopy (TEM) images revealed that the mycelium underwent plasmolysis in the medium (Fig. 4C). Statistical analysis showed that more than 30% of ΔtagG mycelia

showed plasmolysis, while less than 5% of WT mycelia did ($n > 200$). The fermentation results showed that the growth curve of the $\Delta tagG$ strain was not significantly different from that of the WT, but the yield of daptomycin was lower (Fig. 4D and E). The *SrtagG* complementation experiment confirmed that the phenotypic changes in $\Delta tagR$ were caused by *SrtagG* disruption (Fig. S3B, D, and E). In summary, the observed phenotype of the $\Delta tagG$ strain, particularly the delayed differentiation and plasmolysis, was consistent with reported studies.

The next step was to perform the functional complementation experiment in $\Delta tagG$ with the homologous gene of *tagR*. Studies have confirmed that BsTagG is a WTA ABC transporter permease that can transport WTA in different genera, even if the WTA main chain is different (46). As mentioned earlier, *SrtagG* was predicted to encode a WTA ABC transporter permease. Therefore, *BstagG* was complemented into $\Delta tagG$, and the genotype verification of the complemented strain *S. roseosporus* $\Delta tagG::BstagG$ is shown in Fig. S7C. If $\Delta tagG$ WTA-deficient phenotype is rescued by *BstagG*, then *SrtagG* should have the same function as *BstagG*. The results showed that the phenotype of the complemented strain $\Delta tagG::BstagG$ was similar to that of the WT in terms of mycelial growth, sporulation ability, and recovery of the mycelial surface roughness, as observed by SEM, and recovery of plasmolysis as observed by TEM (Fig. 4A to C). Compared with that of the $\Delta tagR$ strain, the yield of daptomycin from the $\Delta tagG::BstagG$ strain also showed a relative recovery (Fig. 4D). *BstagG* effectively complemented the function of *SrtagG*, suggesting that *SrtagG* has the same function as *BstagG*.

Moreover, in the genome of *S. roseosporus*, local BLAST results revealed that SrtagG and SrtagH had a pair of alleles, orf659 and orf658, with identities of 60.53% and 73.36%, respectively (Fig. S8A). According to the results of transcriptome analysis and reverse transcriptase quantitative PCR (RT-qPCR), the expression levels of these two genes were not affected by the change in the expression levels of their alleles *SrtagG* and *SrtagH* (Fig. S8B and C). The expression of *orf658* and *orf659* was stable across multiple genotypes, suggesting that these genes are constitutively expressed.

To briefly summarize the TagR-WTA transporter regulatory pathway, TagR and SroLm3 enhanced the expression of *tagR*, and TagR negatively regulated its target genes WTA ABC transporters *SrtagG* and *SrtagH* (Fig. 5A).

**Enhancing environmental stress resistance and the yield of daptomycin due to *tagR* deletion.** Maintenance of the ionic balance is the primary protective mechanism of microbial cells against hyperosmotic stress (1). According to previous studies, to tolerate high osmotic pressure, bacteria can adjust their cell wall cation-binding capacity by adjusting the WTA level (44, 47, 48) (Fig. 5A). TagG is a WTA transporter, and the $\Delta tagG$ mutant exhibited plasmolysis in TEM images. Therefore, WTA transporter deficiency leads to a decrease in tolerance to high osmotic pressure. Because the WTA ABC transporter was highly expressed in the $\Delta tagR$ strain, we conjectured that this strain might have a higher osmotic tolerance; therefore, the osmotic tolerance of the $\Delta tagR$ and WT strains was tested. The $\Delta tagR$ strain showed higher tolerance to a high concentration of $Mg^{2+}$, which was most obvious at 400 mM $MgCl_2$ (Fig. 5B), and the results were consistent with expectations.

Much evidence supports the theory that negatively charged phosphodiester groups in WTAs serve to stabilize cation gradients and buffer the cell membrane by interacting with mobile cations in the cell wall and the region between the cell wall and membrane (49–51). However, high-yield fermentation of daptomycin requires supplementation with DA, which is toxic to cells. One of the mechanisms of its toxicity involves the impairment of electron transfer in the electron transport chain, thereby reducing the proton gradient and membrane potential and leading to a reduction in ATP production, thus causing bacteria to lose an important source of energy, and ultimately inhibiting cell growth (7, 52). Therefore, we conjectured that stabilization of the cation gradient by WTA might improve the tolerance to DA and subsequently enhance the yield of daptomycin (Fig. 5A). Thus, fermentation with gradient addition of DA was performed using the WT and $\Delta tagR$ strains to investigate this hypothesis (Fig. 5C and D). As the DA feeding medium (DA:methyl oleate = 1:1) is both a precursor and a carbon source,

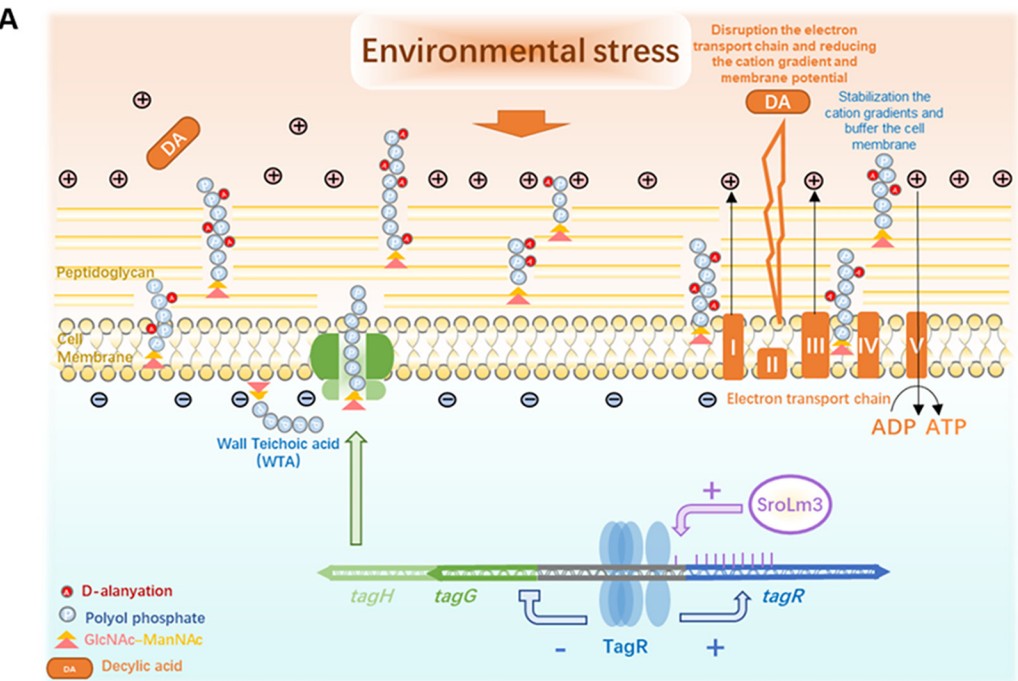

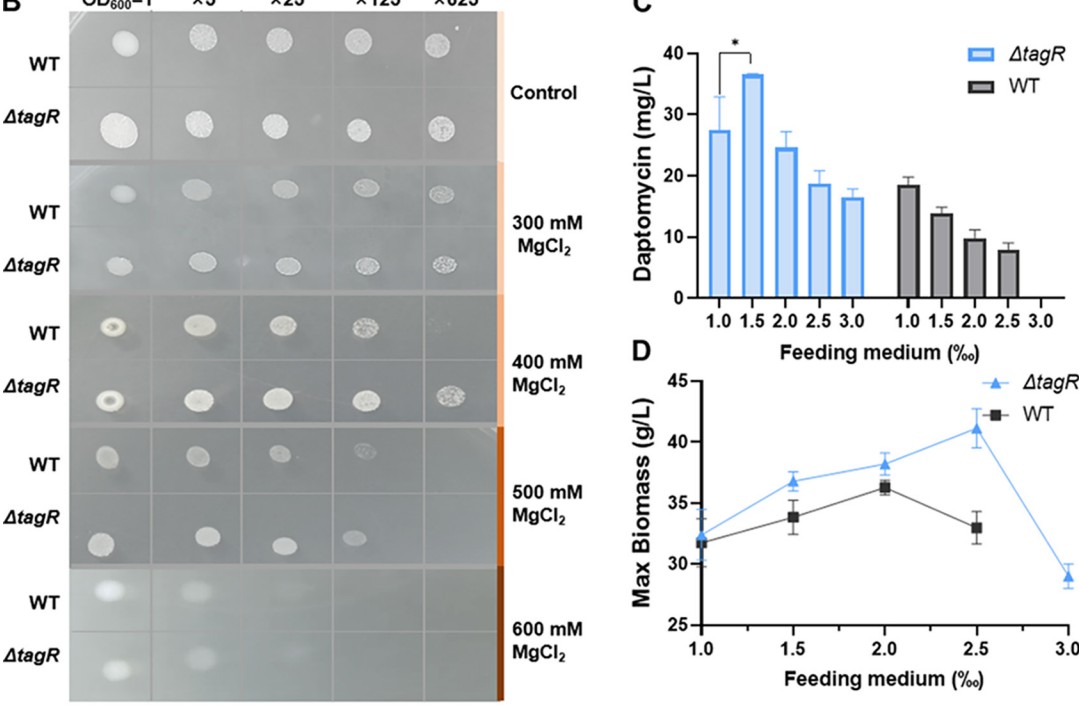

**FIG 5** Environmental stress resistance of WT and Δ*tagR*. (A) Schematic diagram of Δ*tagR* environmental stress resistance. SrTagR and SroLm3 could enhance *SrtagR* expression, and SrTagR negatively regulated the expression of the WTA ABC transporters *SrtagG* and *SrtagH*. Knockout of *tagR* enhanced the expression of WTA transporters and might increase the cell wall cation-binding capacity to allow cells to tolerate high cation concentrations and DA. (B) Comparison of osmotic pressure tolerance of WT and Δ*tagR*. The first horizontal row indicates the initial bacterial concentration and dilution ratio. (C) Yields of daptomycin from WT and Δ*tagR* with the addition of different volumes of DA feeding medium (DA:methyl oleate = 1:1) ($n$ = 3, mean with SD). (D) The maximum biomass of the WT and Δ*tagR* strains with the addition of different volumes of DA feeding medium ($n$ = 3, mean with SD). $P <$ 0.05 was summarized with 1 asterisk [*].

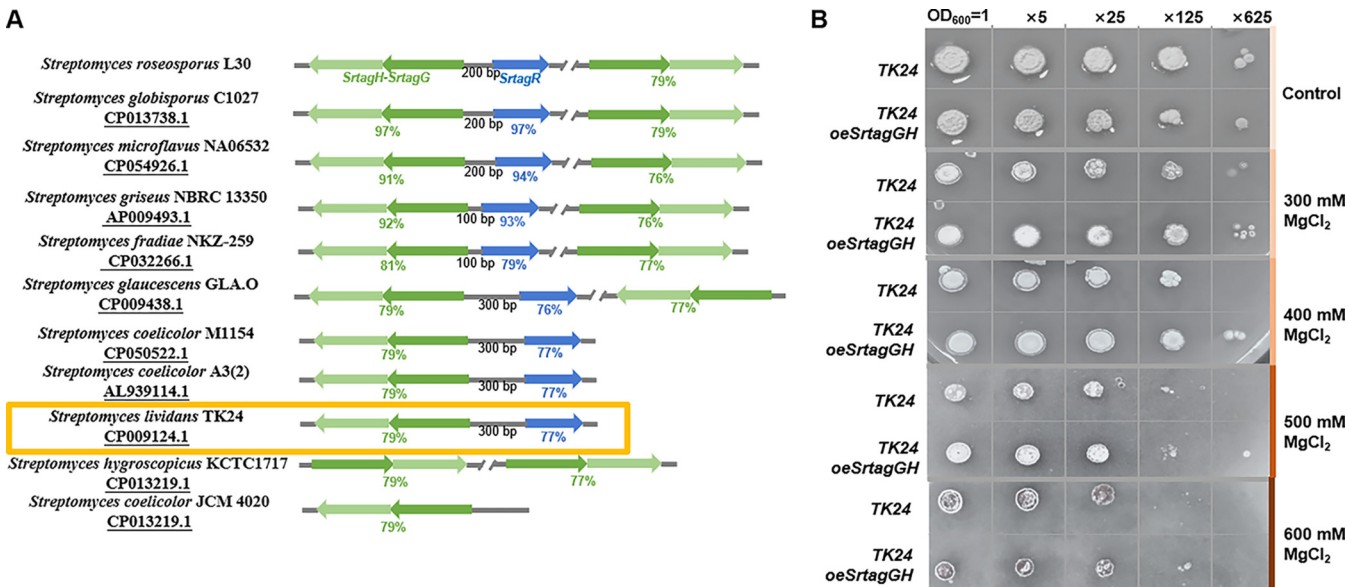

**FIG 6** WTA transferase and its regulator in *Streptomyces* species and osmotic pressure tolerance analysis of *S. lividans*. (A) Schematic diagram of the distribution of WTA transferase and its regulator in *Streptomyces* spp. genomes. The GenBank number of the genome is shown in black. The green arrows indicate the homology to the *SrtagH-SrtagG* fragment, and the blue arrows indicate the homology to the *SrtagR* fragment. (B) Comparison of osmotic pressure tolerance of *TK24* and *TK24oeSrtagGH*. The first horizontal row indicates the initial bacterial concentration and dilution ratio.

increased biomass can strongly indicate the increased tolerance to DA. With the addition of 2 ‰ and 2.5 ‰ DA to the Δ*tagR* strain and 1.5 ‰ and 2 ‰ DA to the WT, the biomass increased, but the yield of daptomycin conversely decreased. This finding indicated that the secondary metabolism was affected in these strains. Furthermore, in both the Δ*tagR* and WT strains, growth inhibition was obvious when large amounts of DA were added, so the biomass is a more believable indicator. The maximum biomass of the WT was achieved with a 2.0‰ addition of DA feeding medium, whereas the maximum biomass of Δ*tagR* was achieved with a 2.5‰ addition. The maximum yield of WT was achieved with a 1‰ addition, whereas that of Δ*tagR* was achieved with a 1.5‰ addition; the latter represented a 100% higher yield of daptomycin than that of the former (Fig. 5C). In short, fermentation with the gradient addition of DA showed that Δ*tagR* tolerated more DA than WT, as indicated by the daptomycin yield and the maximum dry weight.

**Role of TagR target genes in other *Streptomyces* spp.** To further explore the functions of *SrtagR*, *SrtagG*, and *SrtagH* (*SrtagGH*), we investigated the arrangement of the WTA ABC transporter and its regulatory proteins in the genomes of typical model strains and antibiotic-producing *Streptomyces*. The results showed that genes with high homology (76%) to *SrtagR* fragments and *SrtagGH* fragments existed in most of the searched strains, which indicated that the TagR-WTA transporter regulatory pathway might be universally present in *Streptomyces* (Fig. 6A). However, the distances and copy numbers of the genes differed. There were one or two copies of *SrtagGH* alleles in the genome of each strain. The distance between the *SrtagR* homologous fragments and *SrtagGH* homologous fragments varied from 100 to 300 bp. Some strains, such as *Streptomyces hygroscopicus* KCTC1717, did not have a regulator in the vicinity of the *SrtagGH* homologous fragments. This situation may imply the absence of specific regulators such as TagR. Moreover, differences existed among different subspecies. For example, unlike *S. coelicolor* M1154 and A3 (2), the JCM 4020 strain had no *SrtagR* homologous fragment in the vicinity of the *SrtagGH* homologous fragment. In short, the WTA ABC transporter genes and their regulator gene *tagR* have diverse arrangements among *Streptomyces* spp. Recently, the same phenomenon for WTA gene cassettes was reported in *B. subtilis* (53). The diverse arrangement implies that the intensity of regulation of the WTA transporter may differ among strains, conferring different cell wall polyanionic matrices and determining the cationic binding capacity (54). By naturally rearranging their WTA cassettes, strains may be able to occupy diverse niches (53). Similarly,

we may be able to exploit the TagR-WTA regulatory pathway to engineer strains with enhanced environmental stress resistance.

The premise for the widespread application of the regulatory pathway is that the effectiveness of the WTA transporter in improving environmental adaptability is proven in other strains. To investigate whether an enhanced WTA transport capacity could also improve osmotic stress resistance in other strains, the model strain *Streptomyces lividans TK24* was selected for further research. An extra copy of the WTA transporter was highly expressed to determine whether its antihypertonic ability was enhanced. The high-expression plasmid pIJ8661-*SrtagGH* was transferred into strain *TK24* by conjugation, and the genotype and *SrtagGH* expression of *S. lividans TK24oeSrtagGH* are shown in Fig. S9A and B. Osmotic tolerance tests showed that *TK24oeSrtagGH* exhibited a better tolerance, and the difference was most evident at 400 mM $MgCl_2$. This result indicated that increased WTA transport ability in other *Streptomyces* spp. could also improve strain hyperosmotic tolerance. The TagR-WTA transporter regulatory pathway has the potential for wide application for hyperosmotic tolerance improvement. *Streptomyces* spp. are important producers of microbial drugs, and this improvement is undoubtedly beneficial for the production efficiency and fermentation processes.

## DISCUSSION

DNA methylation plays an important role in cell defense and gene regulation (55). Based on the DNA methylome and KEGG pathway assignments in *S. roseosporus*, we discovered an environmental stress resistance regulator, TagR. Further study identified the target genes of TagR as the WTA ABC transport system and its negative regulation. The *tagR* transcription was found to be positively self-regulated and regulated by methylation modification, and TagR binding sites on the *tagG* promoter were determined. Furthermore, mechanistic analysis and experimental results showed that the knockout of *tagR* could effectively improve the strain resistance to hyperosmotic stress and DA. The Δ*tagR* mutant showed 100% higher daptomycin yield than the WT with a 50% higher DA addition. Moreover, increased expression of the WTA transporter in the model strain *S. lividans TK24* improved its resistance to hypertonia. The TagR-WTA transporter regulatory pathway, which may be universally present in *Streptomyces* spp., has the potential for wide application for hyperosmotic tolerance improvement.

WTA is one of the major components of the Gram-positive cell wall and one of the factors that mediate the interactions between cells and environmental factors. WTA has essential physiological functions in cell division, drug resistance, and biofilm formation (56–58). Moreover, it has been reported that the yield of different secreted products could be improved by altering WTA degradation in *Escherichia* and changing the D-alanylation modification of WTA in *Bacillus* (59–61). However, in actinomycetes the mechanisms of WTA synthesis and regulation are poorly understood (54). Herein, the TagR-WTA transporter regulatory pathway was identified for the first time, and the diversity of its gene arrangement in the genome was also discussed. This study enriches our understanding of the environmental adaptability and WTA transport in actinomycetes.

The discovery and characterization of regulators provide a theoretical basis for the optimization of industrial actinomycetes. Conventional mining strategies are mainly based on the level of transcription. At present, DNA methylome analysis is very rarely used to explore the regulators of environmental stress resistance. The widespread use of such studies will promote the discovery of key regulators, contributing to theoretical innovations in strain optimization. In this study, the TagR-WTA transporter regulatory pathway identified by this strategy is expected to be a new target for improving hypertonic tolerance. Moreover, it is worth digging into how the methyltransferase SroLm3 regulates TagR expression. Based on the experimental results, a reasonable hypothesis is that the m4C methylation introduced with in the *tagR* promoter region by SroLm3 may enhance the activity of TagR, but more experiments are needed to confirm it.

The various environmental stresses faced by strains pose a bottleneck to further improving productivity during fermentation. Current approaches to improving environmental

stress tolerance and exploring its mechanisms have mostly been based on phenotypes, such as adaptive evolution (1). The stress resistance phenotypes are established first, then the mechanisms are analyzed, and finally the analysis results are verified by *in vivo* or *in vitro* experiments (8, 62). Such an approach requires 3 steps. In this study, we first screened for environmental stress resistance regulators based on DNA methylome analysis and KEGG pathway assignment followed by functional validation. This approach was used to perform analyses from the genotype to phenotype and required only two steps, mining and validation, which is more efficient than other approaches. With the rapid development of omics and bioinformatics approaches, gene function prediction has become increasingly more accurate, which enables greater certainty during gene mining and validation. With its wide application, it is expected to rapidly identify the key targets of stress resistance and improve the efficiency of industrial strain optimization.

Herein, the DNA methylome was innovatively used to mine regulators of environmental stress resistance in actinomycetes, and a new regulator, TagR, was discovered. Further analysis revealed the regulatory mechanisms of TagR, leading to its application for improving environmental stress resistance and antibiotic yield. This study provides a new perspective on and a new feasible scheme for the optimization of industrial actinomycetes.

## MATERIALS AND METHODS

**Strains, plasmids, primers, and culture conditions.** All strains used in this research are listed in Table S1. All plasmids used are listed in Table S2. All primers are provided in Table S3. The media and culture conditions used in this study are described in the Supplemental Materials and Methods. The processes of in-frame deletion (Fig. S1A to C) and construction of an overexpressing strain are described in the Supplemental Materials and Methods. The plasmids pKC1139 (63) and pIJ8661 (64) were used in the construction processes.

**RT-qPCR and transcriptome analyses of *S. roseosporus*.** The primers used in the RT-qPCR analysis are listed in Table S3, and the RT-qPCR analysis was performed as previously described (65). The sigma factor gene *hrdB* was used as an internal control. RNA extraction, library construction, transcriptome identification, and analysis of WT and Δ*tagR* strains were performed by Azenta (Suzhou, China). Detailed procedures are described in the Supplemental Materials and Methods. The transcriptome data have been deposited in the Sequence Read Archive (SRA), as described in the "Data availability" section below.

KEGG is a collection of databases of genomes, biological pathways, diseases, drugs, and chemical substances (http://en.wikipedia.org/wiki/KEGG). We used scripts in-house to determine the enrichment of significant differential gene expression in KEGG pathways.

**Morphological observation of mutants.** The solid media used for the morphological observation were mentioned above. *S. roseosporus* mutants were cultured on R5 solid medium twice. Spores were collected and streaked on selected solid media. Petri dishes were kept in sealed plastic bags and placed in a bacteriological incubator at 30°C for several days. The growth status was recorded on the 5th and 10th days.

The phenotypes of the mycelia and spores were observed using SEM. Samples were fixed in 0.1 M phosphate buffer (pH 7.4) (PBS) with 2.5% glutaraldehyde for more than 8 h. Then, samples were washed 3 times with PBS and fixed with 1% $OsO_4$ for 2 h. Next, $OsO_4$ was removed and the samples were washed 3 times with PBS. Different gradients of ethanol solutions (50%, 70%, 80%, 90%, and 100%) for dehydration were added for 15 min each. Finally, the samples were dried through the critical point, and coating and SEM observations were completed by the staff at the Center of Cryo-Electron Microscopy (CCEM).

TEM sample preparation and observation were completed by the staff at the CCEM. The detailed process is described in the Supplemental Materials and Methods.

**Dry weight determination.** The strains were cultured in TSB seed medium to an optical density at 600 nm ($OD_{600}$) of 0.4 and then transferred to YEME medium. Mycelia were collected at 24, 48, 72, 96, 120, and 144 h and placed in a metal bath at 85°C for 72 h until the water was completely evaporated. The dry weight was measured with an electronic balance (dry weight = sample tube-empty tube; $n = 3$).

**HPLC assay of daptomycin.** The fermentation broth was treated with 3 volumes of methanol and centrifuged. The supernatants were collected and filtered through a Millipore membrane for HPLC analysis. The secondary metabolites were analyzed using a 1260 Infinity II LC System (Agilent Technologies) with the method described in Table S4. Pure daptomycin was used as a standard.

**EMSA and DNase I footprinting assay.** TagR was expressed in *E. coli* BL21 harboring the plasmid pET28a-*tagR* and purified from the soluble fractions (66). The purified protein was added to 10% glycerol, aliquoted, and stored at −80°C. Then, 5′-(6-FAM)-labeled DNA probes were generated by PCR of the plasmids pKC1139 and pKC1139-$P_{tagR}$ with primer M13-F(5′-FAM)/M13R for EMSAs. An independent sequence on pKC1139 was used as a control ($P_{col}$). The probe length in the experiment is $P_{col}$ (278 bp), $P_{tagR(tagG)}$ (182 bp), $P_{orf488}$ (115 bp), $P_{orf492}$ (142 bp), $P_{orf496}$ (129 bp), $P_{orf3563}$ (77 bp), $P_{bind}$ (113 bp), and $P_{muta}$ (113 bp). The 5′-(6-FAM)-labeled mutant probe was designed based on the results of EMSA and

footprinting results and generated by PCR with the primers listed in Table S3. Each binding reaction contained 0.5 ng of the probe and increasing concentrations of TagR as indicated (67).

The 5′-(6-FAM)-labeled DNA probes for the DNase I footprinting assay were generated from the plasmid pKC1139-$P_{tagR}$ using the primer $tagGR$-P-F(5′-FAM)/$tagGR$-P-R. The DNase I footprinting assay was carried out as previously described (67).

**GusA reporter assays.** A reporter system based on GusA (the $\beta$-glucuronidase enzyme) was constructed and utilized to study the $tagR$ promoter. The primer pair pIJ776-Neo-F/pIJ776-Neo-R was used to amplify $neo$ fragments from the pIJ776 plasmid. The primer pair pSET152-Ep-gusA-F/pSET152-Ep-gusA-R was used to amplify the $gusA$ fragment from the pSET152-Ep-$gusA$ (68) plasmid. The above two DNA fragments were cloned into the pLRM02 (65) plasmid, and the resulting pLRM02-$neo$-$gusA$ plasmid was used as a GusA reporter plasmid. The $tagR$ promoter ($P_{tagR}$) was obtained by PCR of the WT genome with the primers gusA-$tagGR$-P-F/gusA-$tagGR$-P-R. The linearized pLRM02-$neo$-$gusA$ was generated by the digestion with NdeI and BglII. Then, $P_{tagR}$, which contains the 182-bp sequences of Fig. 3C, was inserted into linearized pLRM02-$neo$-$gusA$ (Fig. S6).

The engineered plasmid pLRM02-$neo$-$gusA$-$P_{tagR}$ was transformed into $S. roseosporus$ L30, $S. roseosporus$ $\Delta tagR$, and $S. roseosporus$ $\Delta srolm3\Delta tagR$ via conjugation and then integrated into the genome. The mutants were identified by PCR (data not shown).

The strains were grown in TSB and YEME for 36 h, and mycelia were harvested for subsequent experiments. Spectrophotometric measurement of glucuronidase activity in cell lysates was carried out after 2 h of incubation at 37°C as previously described (69). The protein concentrations in the lysates were measured (Bradford assay) to correct for subtle differences between samples.

**Osmotic stress tolerance experiment.** $S. roseosporus$ and $S. lividans$ were grown at 30°C in TSB to an $OD_{600}$ of 1. Then, $S. roseosporus$ strains were transferred to R5 medium with different $MgCl_2$ concentrations. $S. lividans$ strains were transferred to ISP4 medium with different $MgCl_2$ concentrations. All strains were cultured at 30°C for 5 days.

**Data availability.** The transcriptome data are available in the SRA under accession number PRJNA903398.

## SUPPLEMENTAL MATERIAL

Supplemental material is available online only.

**SUPPLEMENTAL FILE 1**, PDF file, 1.3 MB.

## ACKNOWLEDGMENTS

We thank Azenta Co. (Suzhou, China) for assistance with transcriptome analysis. We thank G.-Z. Zhu from the Center of CCEM, Zhejiang University for her technical assistance with SEM. We thank J.-S. Guo from the Center of CCEM, Zhejiang University for his technical assistance with high-power field sample preparation and TEM.

This work was supported by the Natural Science Foundation of China (grant number 32170057) and the National Key R&D Program of China (grant number 2019YFA09005400).

Y.-Q. Li and W.-L. Gao designed the study. W.-L. Gao conducted the experiments and wrote the manuscript. Y.-Q. Li, J.-L. Fang, and W.-F. Xu revised the manuscript. J.-L. Fang, C.-Y. Zhu, W.-F. Xu, and Z.-Y. Lyu assisted with the experiments (fermentation, construction of plasmids, and analysis of HPLC data). All authors read and approved the final manuscript.

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
