## [Reviewer comments · Microbiology Spectrum]

Microbiology Spectrum

Identification and Characterization of a New Regulator TagR for Environmental Stress Resistance Based on the DNA Methylome of *Streptomyces roseosporus*

Wen-Li Gao, Jiao-Le Fang, Chenyang Zhu, Wei-Feng Xu, Zhong-Yuan Lyu, Xin-Ai Chan, Qing-Wei Zhao, and Yong-Quan Li

Corresponding Author(s): Yong-Quan Li, Zhejiang University School of Medicine

Review Timeline:

Submission Date:	January 24, 2023
Editorial Decision:	March 4, 2023
Revision Received:	March 17, 2023
Accepted:	April 10, 2023

Editor: Wenli Li

Reviewer(s): The reviewers have opted to remain anonymous.

Transaction Report:

DOI: <https://doi.org/10.1128/spectrum.00380-23>

March 4, 2023

Prof. Yong-Quan Li
Zhejiang University School of Medicine
Institute of Pharmaceutical Biotechnology
866# Yuhangtang Road
Hangzhou, Zhejiang Province 310058

Re: Spectrum00380-23 (Discovery and Characterization of a New Regulator TagR for Environmental Stress Resistance Based on the DNA Methylome in *Streptomyces roseosporus*)

Dear Prof. Yong-Quan Li:

Link Not Available

Sincerely,

Wenli Li

Journals Department
Reviewer comments:

Reviewer #1 (Comments for the Author):

Dear authors,

I consider the topic of interest for people working on the optimization of industrial Actinomycetes. However, I have some concerns:

- How many methylated nucleotides were present upstream tagR? The fact that the gene was methylated can be important, given that DNA methylation in actinomycetes has been considered a new mechanism of gene regulation only recently, and needs to be highlighted in the text. However, the choice to knock out tagR derived from different evidence and other works have already found its role in antibiotic biosynthesis. Thus, I'd modify the title. Infact, the clone overexpressing TagR in *E. coli* was

already available from a previous work.

- The word "yield" is in some places incorrectly used, you should specify you refer to antibiotic, for example.
- It is not perfectly clear to me the sentence "strains face environmental stresses during fermentation", considered the references to *Escherichia coli* and yeasts.
- Some references reporting the importance of DNA methylation in the model strain *S. coelicolor* are not reported, i.e. doi: 10.1038/s41598-018-32027-8; doi: 10.1093/nar/gku1376.
- Please, specify the sentence "Here, the specific strategy applied was to screen the regulatory genes based on the difference in methylation sites between WT and Δ sroLm3;" where were the methylation sites? How many? How far from the translation start site?
- It is not perfectly clear to me why the sentence was included "Moreover, in organisms with small genomes, most cis-regulatory elements (CRMS) are close to their target genes (29)."
- The first paragraph describing the choice of tagR is mixed up. Please rephrase it.
- The amplicon sizes should be provided.
- I did not find the size of the probe used in EMSA. Were competitors added into the binding reactions?
- For some experiments, two media were used. Why?
- At what time was the measurement of glucuronidase activity done?
- The sentence "SrTagH is probably related to the transport of WTA, as well as the cotranscribed SrtagG is probably." Should be rephrased.
- The sentence "In addition, local BLAST results revealed that SrtagG and SrtagH had a pair of alleles, orf659 and orf658, in the genome of *S. roseosporus*, with identities of 60.53%, 73.36% " Should be rephrased.
- qRT-PCR analysis was not described in the main text.
- Results are mixed with discussion, but a separate discussion (very poor and repetitive, as a consequence) is also present. I suggest to present results and discussion and then the conclusion of the work.
- The authors should check the English.

Reviewer #2 (Public repository details (Required)):

transcriptomic data is included in the study and has been deposited in NCBI's SRA (PRJNA903398)

Reviewer #2 (Comments for the Author):

The authors present the identification and functional characterization of TagR, a TetR protein involved in regulating the hyperosmotic resistance in *S. roseosporus*. The conclusions inferred are sound and in accordance with the results presented.

The figure legends are minimalist and lack the information needed for proper interpretation. For instance, fig 3B lacks the legend for the chromatogram lines, fig 5A description draws conclusions not presented (enhanced WTA transport), in fig 5B and fig 6b what means x5, x25, etc....,

L113 - At this point is not clear the genomic context of *tagR*. Although it is presented later in the manuscript (Fig 6A), it is important to clarify it at this point, particularly because it is relevant to know that it is located divergently from *tagH* and *tagG*. In addition, it should be included some basic features of *tagR* e.g. length of the coded protein, protein domain architecture, etc..

Fig 3C - It should be included a representation of all promoter elements such as +1 sites of *tagR* and *tagG* as well the putative rbs, -10 and -35 elements of the bidirectional promoter. The information is relevant for the the inducing/repressing mechanism of TagR and how it could influence the RNA polymerase activity.

Throughout the text there are misleading statements regarding the role of SroLm3 in *tagR* transcription (e.g. lines 261 and other places). There is no evidence that SroLm3 binds to the promoter region of *tagR* regulating its transcription directly. Instead, the results suggest that the m4C methylation introduced at the *tagR* promoter region by SroLm3 enhances the activity of TagR regarding its own expression. This point should be clear on the manuscript.

The production of daptomycin was assessed by supplementing the culture broths of wt and Δ tagR strains with DA. The values are presented as volumetric production. However, it is clear that the addition of DA affects the biomass production. To better show the influence of DA, daptomycin specific production should be presented.

- 176: The meaning of the DA abbreviation should be included at least once in the main text. It is only mentioned in the abstract and fig 5 legend

-1147: stationary instead of stable

- L190: TagR instead of tagR

- L190 - 192: The text suggests that 2 promoters (Pneo and PgusA) are replaced by PtagR in the pRM02 based construction. However, in Fig S6 it seems that both genes, neo and gusA, are co-transcribed from one promoter. Please clarify. In addition, it should be clearly indicated what was the PtagR region cloned (as presented in Fig3B)

- Throughout the text replace qRT-PCR by RT-qPCR. In addition, although the procedure for RT-qPCR is referred to previous

- work, it should be indicated the gene used as reference as well as its expression stability (e.g the M value)
- L216: incomplete/confuse sentence.
 - L242: incomplete/confuse sentence. "complementing the Δ tagR with an heterologous gene with known function"
 - L307 - 308: Did those strains had a regulator elsewhere or no TagR homologous was found? Please clarify.
 - L336, I354: tagR transcription
 - I368: transporter in Actinomycetes
 - Remove panel E from fig 3. It is repeated in Fig 5A.

Staff Comments:

Preparing Revision Guidelines

Please return the manuscript within 60 days; if you cannot complete the modification within this time period, please contact me. If you do not wish to modify the manuscript and prefer to submit it to another journal, please notify me of your decision immediately so that the manuscript may be formally withdrawn from consideration by Microbiology Spectrum.

Dear editor and reviewers:

Thank you very much for your comments and professional advice. Your suggestions have made our manuscript more rigorous. We carefully revised the manuscript and highlighted the revisions in yellow. And we have done some polishing on my English, corrected some clerical mistakes and highlighted them in cyan. We hope this revised manuscript has addressed the concerns.

Responses to Reviewer 1

Point 1: How many methylated nucleotides were present upstream *tagR*? The fact that the gene was methylated can be important, given that DNA methylation in actinomycetes has been considered a new mechanism of gene regulation only recently, and needs to be highlighted in the text. However, the choice to knock out *tagR* derived from different evidence and other works have already found its role in antibiotic biosynthesis. Thus, I'd modify the title. Infact, the clone overexpressing TagR in *E. coli* was already available from a previous work.

Response 1: We appreciate it very much for this good suggestion. We have added the description of the methylated nucleotides in *tagR* in the penultimate sentence of the first paragraph (L117) as follows:

-L117 “The length of *tagR* was found to be 621 bp, and it encoded 206 amino acids. In WT, cytosine was modified at 9 bp upstream of *tagR*, and there were 9 cytosine bases modified in the CDS region, which were at positions 213, 237, 309, 314, 423, 462, 476, 477, and 554. All these modifications disappeared in the *ΔsroLm3* mutant.”

Moreover, the title has been changed to “Identification and Characterization of a New Regulator TagR for Environmental Stress Resistance Based on the DNA Methylome of *Streptomyces roseosporus*”.

Point 2: The word "yield" is in some places incorrectly used, you should specify you refer to antibiotic, for example.

Response 2: Thank you for your suggestions. It has been revised in the Introduction and the last paragraph of Importance as follows:

-L41 “The TagR-WTA transporter regulatory pathway improved the resistance and antibiotic yield of strains and has the potential for wide application.”

-L91 “Further analysis revealed the mechanisms of TagR involvement, leading to its application to improve strain resistance and the yield of daptomycin.”

Point 3: It is not perfectly clear to me the sentence "strains face environmental stresses during fermentation", considered the references to *Escherichia coli* and yeasts.

Response 3: Thank you for your questions. We have rephrased the sentence in the appropriate place (L49) as follows:

-L49 “Many microorganisms, including actinomycetes, generally face environmental stresses during fermentation.”

Point 4: Some references reporting the importance of DNA methylation in the model strain *S. coelicolor* are not reported, i.e. doi: 10.1038/s41598-018-32027-8; doi: 10.1093/nar/gku1376.

Response 4: As suggested, we have added more references in the second paragraph of the Introduction (L62 and L70) as follows:

-L62 “DNA methylation not only helps proteins recognize alien DNA fragments, such as Sco5333 from *Streptomyces coelicolor* and Tbis1 from *Thermobispora bispora* recognize m5C, but also regulates gene expression.”

-L70 “Cytosine methylation controls morphophysiological differentiation and actinorhodin production in *S. coelicolor*.”

Point 5: Please, specify the sentence "Here, the specific strategy applied was to screen the regulatory genes based on the difference in methylation sites between WT and Δ sroLm3;" where were the methylation sites? How many? How far from the translation start site?

Response 5: Thanks for your great suggestions on improving the accessibility of our manuscript. We have provided a brief description in the first paragraph of the Result (L101). Our research group has previously published an article on methylation sequencing, which has been cited in this paper, more methylome data can be found in the citation. Here is the link to the data, <https://www.ncbi.nlm.nih.gov/bioproject/PRJNA758636/>.

-L101 “There were 23,847 m4C modification sites in WT and 15,646 in *ΔsroLm3* and there were 552 presumed regulatory genes in the genome. Most of the modifications were in the CDS region, and therefore, regulatory genes were screened according to the modification differences in this region.”

Point 6: It is not perfectly clear to me why the sentence was included "Moreover, in organisms with small genomes, most cis-regulatory elements (CRMS) are close to their target genes (29)."

Response 6: Thank you for your question. Since the regulator may be close to its target gene, we would select regulators in the regions with a high density of genes in environmental information processing pathways. However, too much explanation may add confusion, we have deleted this sentence.

Point 7: The first paragraph describing the choice of *tagR* is mixed up. Please rephrase it.

Response 7: Thank you for your suggestion. We have divided the first paragraph into two, simplified the expression, and added relevant information on methylation sites in the genome and *tagR*, hoping to make it clear (L95-L121).

Point 8: The amplicon sizes should be provided.

Response 8: Thanks for your suggestion on improving our manuscript. I have added this information to Table S3.

Point 9: I did not find the size of the probe used in EMSA. Were competitors added into the binding reactions?

Response 9: Thank you for your question. We have added the information in the method section(L443) as follows:

-L443 “An independent sequence on pKC1139 is used as control (P_{col}). The probe length in the experiment is P_{col} (278 bp), $P_{tagR(tagG)}$ (182 bp), P_{orf488} (115 bp), P_{orf492} (142 bp), P_{orf496} (129 bp), $P_{orf3563}$ (77 bp), P_{bind} (113 bp), P_{muta} (113 bp).”

The control probe for the EMSA, P_{col} , was an unrelated fragment from the pKC1139 plasmid, and no competitor was used. Moreover, the mutated sequence was assay controlling with the original sequence, and did not use competitors.

Point 10: For some experiments, two media were used. Why?

Response 10: Thank you for your detailed review. In the “Osmotic stress tolerance experiment” subsection, different solid media were needed to cultivate different strains. *S. roseosporus* was cultured in R5, while *S. lividans* was cultured in ISP4. In addition, to make the results more solid and reliable, TSB seed medium and YEME fermentation medium were used to cultivate mycelia in GusA reporter analysis.

Point 11: At what time was the measurement of glucuronidase activity done?

Response 11: Thank you for your question. We are sorry for our negligence. The information has been added in the 3rd paragraph of the “GusA reporter analysis” subsection (L468) as follows:

-L468 “Spectrophotometric measurement of glucuronidase activity in cell lysates was carried out after 2 h of incubation at 37°C as previously described (69)”.

Point 12: The sentence "SrTagH is probably related to the transport of WTA, as well as the cotranscribed SrtagG is probably." Should be rephrased.

Response 12: Thanks for your comment on improving the accessibility of our manuscript. We have rephrased and modified the corresponding sentence as follows:

-L230 “SrTagH is probably related to the transport of WTA, and we speculate that its cotranscriptional gene encoded protein, SrtagG, may also involve in WTA transport.”

Point 13: The sentence "In addition, local BLAST results revealed that SrtagG and SrtagH had a pair of alleles, orf659 and orf658, in the genome of *S. roseosporus*, with identities of 60.53%, 73.36%, respectively " Should be rephrased.

Response 13: Thank you for your comment. We have rephrased and modified the corresponding sentence as follows:

-L268 “Moreover, in the genome of *S. roseosporus*, local BLAST results revealed that SrtagG and SrtagH had a pair of alleles, *orf659* and *orf658*, with identities of 60.53% and 73.36%, respectively.”

Point 14: qRT-PCR analysis was not described in the main text.

Response 14: Thank you for your comment. In the first paragraph of the “RT-qPCR analysis and transcriptome analysis of *S. roseosporus*” subsection, the reference ((65), DOI:

10.1007/s00253-018-9103-5) was provided to a previously described method to avoid repetition. Here, we briefly introduce the method that was used.

Mycelia of *S. roseosporus* and its derivatives were collected from fermentation media every 24 h and immediately frozen for RNA extraction. Total RNA was extracted with TRIzol (Sangon) according to the manufacturer's instructions. Genomic DNA was removed with RNase-free DNase I (TaKaRa). cDNA was synthesized with MMLV reverse transcriptase according to the protocol of the manufacturer (TaKaRa). RT-qPCR was performed using SYBR PremixEx Taq II (TaKaRa). The sigma factor gene *hrdB* was used as an internal control for the RT-qPCR assay in *S. roseosporus*. Each RT-qPCR was performed in triplicate. Fold changes in the gene expression levels were calculated using the $2^{-\Delta\Delta Ct}$ method according to the manufacturer's protocol (TaKaRa).

Point 15: Results are mixed with discussion, but a separate discussion (very poor and repetitive, as a consequence) is also present. I suggest to present results and discussion and then the conclusion of the work.

Response 15: Thanks for your comment on improving our manuscript. We have recognized the problem of repetition. The discussion in the Results section was to analyze the data to make reasonable assumptions, which helps to clarify the next steps. After careful consideration, to make the Results section more logically coherent, we would like to include some necessary discussion in the Results section. To be concise and avoid repetition, we have removed repetitive information from the Discussion(L359-L391) and refined the statement (L365).

-L365 “Herein, the TagR-WTA transporter regulatory pathway was identified for the first time, and the diversity of its gene arrangement in the genome was also discussed. This study enriches our understanding of the environmental adaptability and WTA transport in actinomycetes.”

Point 16: The authors should check the English.

Response 16: The paper has been carefully revised. And we polished the English, corrected some clerical mistakes and highlighted them in cyan. Thank you for your suggestion.

This document certifies that the manuscript

Identification and Characterization of a New Regulator TagR for Environmental Stress Resistance Based on the DNA Methylome of *Streptomyces roseosporus*

prepared by the authors

Wen-Li Gao, Jiao-Le Fang, Chen-Yang Zhu, Wei-Feng Xu, Zhong-Yuan Lyu, Xin-Ai Chan, Qing-Wei Zhao, Yong-Quan Li

was edited for proper English language, grammar, punctuation, spelling, and overall style by one or more of the highly qualified native English speaking editors at AJE.

This certificate was issued on **March 13, 2023** and may be verified on the [AJE website](https://www.aje.com) using the verification code **258C-DE7D-7249-99CD-6B6P**.

Neither the research content nor the authors' intentions were altered in any way during the editing process. Documents receiving this certification should be English-ready for publication; however, the author has the ability to accept or reject our suggestions and changes. To verify the final AJE edited version, please visit our verification page at [aje.com/certificate](https://www.aje.com/certificate). If you have any questions or concerns about this edited document, please contact AJE at support@aje.com.

Responses to Reviewer 2

Point 1: The figure legends are minimalist and lack the information needed for proper interpretation. For instance, fig 3B lacks the legend for the chromatogram lines, fig 5A description draws conclusions not presented (enhanced WTA transport), in fig 5B and fig 6b what means x5, x25, etc....,

Response 1: We are sorry for our negligence and have revised the figure and their legends. We have added an annotation to Fig. 3B and a description for chromatogram lines to the legend (L711) as follows:

-L711 “The blue chromatogram line is the group without TagR, and the red chromatogram line is the group with the addition of 10 µg of TagR.”

We have revised the legend to Fig. 5A (L726), as follows:

-L726 “Knockout of *tagR* enhanced the expression of WTA transporters and might increase the cell wall cation-binding capacity to allow the cells to tolerate high cation concentrations and DA.”

We have added the following description to the legends of Fig. 5B and Fig. 6B: “The first horizontal row indicates the initial bacterial concentration and dilution ratio.” (L730 and L742).

Point 2: L113 - At this point is not clear the genomic context of *tagR*. Although it is presented later in the manuscript (Fig 6A), it is important to clarify it at this point, particularly because it is relevant to know that it is located divergently from *tagH* and *tagG*. In addition, it should be included some basic features or *tagR* e.g. length of the coded protein, protein domain architecture, etc.

Response 2: Thanks for your great suggestion on improving the accessibility of our manuscript. We have added a description of the genomic context and protein domains of *tagR* (L122) and have also added the length of the gene and that of its encoded protein, at the end of the previous paragraph (L117):

-L122 “The upstream gene of *tagR* was annotated as an ABC transporter, and its downstream gene was a hypothetical protein, both of which had different transcriptional directions from

tagR. TagR has typical TetR structures, including an N-terminal HTH DNA-binding motif and a C-terminal ligand recognition domain.”

-L117 “The length of *tagR* was found to be 621 bp, it encoded 206 amino acids.”

Point 3: Fig 3C - It should be included a representation of all promoter elements such as +1 sites of *tagR* and *tagG* as well the putative rbs, -10 and -35 elements of the bidirectional promoter. The information is relevant for the the inducing/repressing mechanism of TagR and how it could influence the RNA polymerase activity.

Response 3: We appreciate it very much for this good suggestion. In Fig. 3C, we have added transcription directions of *tagR* and *tagG*, putative -10 and -35 elements, and putative RBSs. The legend has been revised accordingly (L715).

-L715 “The putative -10 and -35 sites of *tagR* are marked in blue, and the putative -10 and -35 sites of *tagG* are marked in green. The putative RBSs are circled with a black dotted line.”

Point 4: Throughout the text there are misleading statements regarding the role of SroLm3 in *tagR* transcription (e.g. lines 261 and other places). There is no evidence that SroLm3 binds to the promoter region of *tagR* regulating its transcription directly. Instead, the results suggest that the m4C methylation introduced at the *tagR* promoter region by SroLm3 enhances the activity of TagR regarding its own expression. This point should be clear on the manuscript.

Response 4: Thank you for your suggestions. To make this clear, I have added the following sentence to the last paragraph of the subsection “Regulatory mechanism of *tagR*” (L214): “However, it is not clear whether this regulation is direct or indirect.” We explained this again in detail in the third paragraph of the Discussion section (L375) as follows:

-L375 “It's worth digging into how the methyltransferase SroLm3 regulates TagR expression. Based on the experimental results, a reasonable hypothesis is that the m4C methylation introduced with in the *tagR* promoter region by SroLm3 may enhance the activity of TagR, but more experiments are needed to confirm it.”

Point 5: The production of daptomycin was assessed by supplementing the culture broths of wt and Δ *tagR* strains with DA. The values are presented as volumetric production. However,

it is clear that the addition of DA affects the biomass production. To better show the influence of DA, daptomycin specific production should be presented.

Response 5: We are so grateful for your suggestion. DA is important for daptomycin production. Daptomycin is a cyclic lipopeptide antibiotic isolated from *Streptomyces roseosporus*. Its N-terminus is a 10-carbon chain (shown in orange below). At the early stage of discovery, daptomycin was synthesized by acylation of A21978C (produced by *S. roseosporus*) and decanoic acid (DA) in vitro (DOI:10.1016/0168-1656(88)90040-5). To date, the fermentation of daptomycin critically depends on the addition of DA. However, DA is highly toxic to *S. roseosporus*, and it is crucial to develop strategies for increasing DA tolerance.

We have rephrased the penultimate paragraph of the Introduction to emphasize the importance of DA (L80). In addition, the description of the fermentation conditions has been improved in the “Media and cultural conditions” section of the supplementary file as follows:
-L79 “The addition of decanoic acid (DA) to the culture broth was shown to be essential for increasing daptomycin yield and productivity.”

“*S. roseosporus* was cultured in TSB at 280 rpm for 36 h, and then mycelia were transferred into YEME at a 1% ratio. Starting from 72 h, every 12 h, 1‰ decanoic acid feeding medium (decanoic acid: methyl oleate =1:1) was added to the fermentation broth (YEME) to promote the synthesis of daptomycin.”

Point 6: - 176: The meaning of the DA abbreviation should be included at least once in the main text. It is only mentioned in the abstract and fig 5 legend.

Response 6: Thank you for your suggestion. We have added this information to the penultimate paragraph of the Introduction (L79) as follows:

-L79 “The addition of decanoic acid (DA) to the culture broth was shown to be essential for increasing daptomycin yield and productivity(30).”

Point 7: -1147: stationary instead of stable

Response 7: Thank you for the detailed review. We have made the corresponding revision (L159) as follows:

-L159 “Based on the growth curves, transcriptome sequencing was performed at 48 h (log phase), 72 h (late log phase), and 120 h (stationary growth phase).”

Point 8: - L190: TagR instead of tagR

Response 8: We are very sorry for our careless mistake and have made the corresponding revision (L201) as follows:

-L201 “To evaluate the possible self-regulation and DNA methylation-based regulation of TagR, the *neo* (kanamycin resistance gene) and *gusA* reporter gene were expressed in tandem, and their common promoter was replaced with P_{tagR} to generate the GusA reporter system (Fig. S6).”

Point 9: - L190 - 192: The text suggests that 2 promoters (P_{neo} and P_{gusA}) are replaced by P_{tagR} in the pRM02 based construction. However, in Fig S6 it seems that both genes, *neo* and *gusA*, are co-transcribed from one promoter. Please clarify. In addition, it should be clearly indicated what was the P_{tagR} region cloned (as presented in Fig3B)

Response 9: Thank you for your suggestions. The *neo* and *gusA* genes were expressed in tandem, and thus, they shared a promoter, P_{tagR} . We have rephrased this accordingly (L201). In addition, P_{tagR} contains the entire 182-bp sequence of Fig3B, shown in Fig3C. We have added this information to the “GusA reporter assays” subsection in the Methods section(L461) as follows:

-L201 “To evaluate the possible self-regulation and DNA methylation-based regulation of TagR, the *neo* (kanamycin resistance gene) and *gusA* reporter genes were expressed in tandem, and their shared promoter was replaced with P_{tagR} to generate the GusA reporter system (Fig. S6).”

-L461 “Then, P_{tagR} , which contains the 182-bp sequences of Fig3C, was inserted into linearized pLRM02-*neo-gusA* (Fig. S6).”

Point 10: - Throughout the text replace qRT-PCR by RT-qPCR. In addition, although the procedure for RT-qPCR is referred to previous work, it should be indicated the gene used as reference as well as its expression stability (e.g the M value)

Response 10: Thank you for your professional suggestion. Our RT-qPCR method and internal control gene were consistent with the previous work, but the M value of the internal control gene was not provided. Initially, *hrdB* was identified as the housekeeping sigma factor responsible for the transcription of essential genes in the model strain *Streptomyces coelicolor* A3(2) (DOI: 10.1016/0378-1119(91)90308-x). Furthermore, *hrdB* has been widely used as an internal control in many studies of gene expression in *Streptomyces roseosporus* (DOI: 10.3389/fbioe.2021.618029, DOI: 10.3390/antibiotics11081065, DOI: 10.1007/s11274-020-02909-z). Therefore, it is of reference value to use *hrdB* as the internal control. In our study, RT-qPCR data were consistent with transcriptome results (Fig S8B and C), which also indicated the reliability of *hrdB* as an internal control. Nevertheless, the expression stability of internal control is indeed important and will be investigated in future work. Thank you again for your suggestion. Information on the reference gene has been added to the Methods section (L407) as follows:

-L407 “The sigma factor gene *hrdB* was used as an internal control.”

Point 11: - L216: incomplete/confuse sentence.

Response 11: Thank you for the detailed review. We have rephrased the sentence as follows:

-L227 “Based on the annotation and homology alignment, the function of SrTagH is probably related to the transport of WTA, and we speculate that its cotranscriptional gene encoded protein, SrtagG, may also involve in WTA transport.”

Point 12: -L242: incomplete/confuse sentence. "complementing the $\Delta tagR$ with an heterologous gene with known function"

Response 12: Thank you for the detailed review. We have rephrased this sentence as follows:

-L255 “The next step was to perform the functional complementation experiment in $\Delta tagG$ with the homologous gene of *tagR*.”

Point 13: - L307 - 308: Did those strains had a regulator elsewhere or no TagR homologous was found? Please clarify.

Response 13: We appreciate your suggestions to make the manuscript more rigorous.

TagR belongs to the TerR family, these family genes are especially abundant in *Streptomyces*. Therefore, these strains also contain this family of proteins which have certain homology with TagR.

In this paragraph, I focused on the arrangement of TagR and target genes, so only the genomic context in the vicinity was emphasized. As related functional genes in prokaryotes usually appear in clusters, specific regulatory genes tend to appear in the vicinity of their target genes. Therefore, the absence of regulatory factors may mean that there is no pathway-specific regulation, but it does not rule out that there are global regulators that can regulate it.

We have added explanations in the article to avoid misleading (L321).

-L321 “This situation may imply the absence of specific regulators such as TagR.”

Point 14: - L336, 1354: tagR transcription

Response 14: Thank you for your suggestion. We have made the corresponding revise (L350).

-L350 “The *tagR* transcription was found to be positively regulated by itself and by methylation modification, and TagR binding sites on the *tagG* promoter were determined.”

Point 15: - 1368: transporter in Actinomycetes

Response 15: Thank you for your suggestion. We have deleted this sentence for brevity.

Point 16: - Remove panel E from fig 3. It is repeated in Fig 5A.

Response 16: Thank you for your suggestion. We have removed Fig 3E.

If there are any deficiencies, please do not hesitate to point them out. We look forward to your reply.

Sincerely,

Yong-Quan Li

Li, Yong-Quan, Ph.D

Qiushi Prof. of Zhejiang University

Prof. of Microbial synthetic biology and Microbial Pharmaceutics

Director of Department of Microbiology/Institute of Pharmaceutical Biotechnology, Zhejiang University

April 10, 2023

Prof. Yong-Quan Li
Zhejiang University School of Medicine
Institute of Pharmaceutical Biotechnology
866# Yuhangtang Road
Hangzhou, Zhejiang Province 310058

Re: Spectrum00380-23R1 (Identification and Characterization of a New Regulator TagR for Environmental Stress Resistance Based on the DNA Methylome of *Streptomyces roseosporus*)

Dear Prof. Yong-Quan Li:

Your manuscript has been accepted, and I am forwarding it to the ASM Journals Department for publication. You will be notified when your proofs are ready to be viewed.

Sincerely,

Wenli Li
Editor, Microbiology Spectrum
